# Pregnant women's unmet need to communicate with a health professional during the SARS-CoV-2 pandemic lockdown in France: The Covimater cross-sectional study

Lucia Araujo-Chaveron[1], Alexandra Doncarli[1]*, Catherine Crenn-Hebert[2], Virginie Demiguel[1], Julie Boudet-Berquier[1], Yaya Barry[1], Maria-Eugênia Gomes Do Espirito Santo[1], Andréa Guajardo-Villar[3], Claudie Menguy[1], Anouk Tabaï[4], Karine Wyndels[5], Alexandra Benachi[6,7], Nolwenn Regnault[1]

1 Non-Communicable Diseases and Trauma Division, Santé Publique France, Saint-Maurice, France,
2 Department of Gynecology and Obstetrics, Louis Mourier University Hospital, AP-HP, Colombes, France,
3 Data Processing, Support and Analysis Department, Santé Publique France, Saint-Maurice, France,
4 Alert and Crisis Department, Santé Publique France, Saint-Maurice, France, 5 Santé Publique France, Hauts-de-France Regional Office, Saint-Maurice, France, 6 Division of Obstetrics and Gynecology, Antoine Béclère Hospital, AP-HP, Clamart, France, 7 Paris Saclay University, Clamart, France

☯ These authors contributed equally to this work.
* alexandra.doncarli@santepubliquefrance.fr

**Data Availability Statement:** Data cannot be shared publicly due to privacy or ethical

## Abstract

During the severe acute respiratory syndrome coronavirus 2 (SARS-CoV-2) pandemic lockdown, communication between pregnant women and health professionals may have become complicated due to restrictions on movement and saturated health services. This could have impacts on pregnancy monitoring and women's wellbeing. We aimed to i) describe the unmet need of pregnant women living in France to communicate with health professionals about the pandemic and their pregnancy during the lockdown, ii) assess the socio-demographic, medical and contextual factors associated with this unmet need. The Covimater cross-sectional study, conducted in July 2020, includes data on 500 adult women's experiences of pregnancy during the first lockdown period in France (i.e., from March to May 2020). The women, all residents in metropolitan France, answered a web-based questionnaire about their conversations with health professionals during the lockdown, as well as their social and medical characteristics. A robust variance Poisson regression model was used to estimate crude or adjusted prevalence ratios (aPRs) for their unmet need to communicate with health professionals about the pandemic and their pregnancy. Forty-one percent of participants reported an unmet need to communicate with a health professional during the lockdown, mainly about the risk of transmitting SARS-CoV-2 to their baby and the consequences for the latter. Factors associated were: i) being professionally inactive (aPR = 1.58,CI95%[(1.14–2.21]), ii) having an educational level below secondary school diploma (1.38,[1.05,-1.81]), iii) having experienced serious arguments/violence (2.12,[1.28–3.52]), iv) being very worried about the pandemic (1.41,[1.11–1.78]), v) being primiparous (1.36, [1.06–1.74]) and vi) having had pregnancy consultations postponed/cancelled by health professionals during the lockdown (1.35,[1.06–1.73]). These results can be used to develop

restrictions. Data are available from the Sante publique France Institutional Data Access (contact: DATA-MAD@santepubliquefrance.fr) for researchers who meet the criteria for access to confidential data.

**Funding:** The author(s) received no specific funding for this work.

**Competing interests:** The authors have declared that no competing interests exist.

targeted strategies that ensure pregnant women are able to i) communicate with health professionals about the potential impact of the SARS-CoV-2 pandemic on their pregnancy, and ii) access up-to-date and reliable information on the consequences of SARS-CoV-2 for themselves and their child.

## Introduction

Data from previous coronavirus epidemics in 2002 (SARS-CoV-1) and 2013 (Middle East respiratory syndrome-related coronavirus, i.e., MERS-CoV) showed that pregnancy was an aggravating factor in respiratory diseases, which in turn are associated with significant maternal-foetal morbidity [1, 2]. Moreover, when the SARS-CoV-2 pandemic began, for months, pregnant women were uncertain about the risk of developing severe forms of COVID-19, the disease caused by SARS-CoV-2, and/or transmitting the virus to their unborn children. Indeed, the first related scientific data published internationally presented contradictory conclusions. While some studies showed an increased risk of complications in infected pregnant women (such as admission to an intensive care unit, invasive ventilation, need for extra corporeal membrane oxygenation, preterm delivery or admission to a neonatal unit for their newborn) [3–5] and a greater possibility of vertical transmission of the virus [6], others found no increased risk compared with infected non-pregnant women [7–9] and no possible transmission of SARS-CoV-2 from mother to foetus [7, 9–12]. These contradictory findings may have generated an increased need for pregnant women to communicate with health professionals about the risks linked to infection (regarding themselves, their foetus or newborn), but also about the course of their pregnancy monitoring or childbirth.

However, such communication may have become more difficult given the spread of the pandemic, the strain it put on healthcare services, and the introduction of lockdown measures. In the United States, a survey conducted between mid-March and mid-May 2020 showed that almost one-third of pregnancy monitoring visits were changed, cancelled or postponed [13]. In France, half (48.9%) of the pregnant women included in the Covimater cross-sectional study reported at least one postponement or cancellation of a consultation or pregnancy check-up during the country's first lockdown (from 17 March to 11 May 2020), whether on their own initiative (23.4%), or that of a hospital/health professional (36.3%) [14]. Another French study reported that 29.5% of pregnancy-related health consultations (mainly healthcare consultations and childbirth preparation sessions) were cancelled during the lockdown [15]. Following these initial cancellations/postponements, recommendations to reorganise and maintain pregnancy monitoring were quickly issued by the French National Authority for Health (HAS, *Haute Autorité de Santé*) [16]. In France, the primary reorganisation strategies were to monitor pregnancies by video or telephone (teleconsultation) and to group certain examinations and consultations [16].

Despite these changes, the overall unprecedented context may have had an impact on the ability of health professionals to respond to their patients' questions. However, even before the pandemic began, several studies highlighted pregnant women's perceived lack of communication from health care professionals about the course of pregnancy [17, 18]. This phenomenon may have increased during the pandemic and the lockdown. In the literature, pregnant women's need to communicate with a health professional during the SARS-CoV-2 pandemic about the risks associated with their own infection or that of their foetus/newborn, about the course of pregnancy monitoring or about childbirth, is poorly documented. Yet descriptive

information on this subject is necessary, particularly to characterise the women most at risk of having unmet communication needs, in order to develop targeted strategies to improve these women's quality of life during pregnancy, a period of great vulnerability in terms of mental health.

In the context of France's first SARS-CoV-2 lockdown, we aimed to (i) estimate the frequency of pregnant women reporting an unmet need to communicate with health professionals, (ii) specify which pregnancy/delivery-related information they were unable to talk about with health professionals, and (iii) assess the factors associated with their unmet need to communicate with health professionals about the impact of the pandemic on their pregnancy/childbirth.

## Materials and methods

### Study design, setting and sample size of Covimater

At our request, a service provider (BVA group) interviewed its unpaid pre-pandemic internet panel of 15,000 future parents or parents of children under 3 years of age in order to create a pseudonymised non-probability sample of 500 pregnant adult women who met the inclusion criteria (described below) and volunteered to participate our survey. Covimater is a cross-sectional study using quota sampling, whereby the study sample was assigned a structure similar to that of the target population (i.e., all pregnant women in France) in order to increase the representativeness. The population of parents of children under 1 year old–as per the National Institute of Statistics and Economic Studies 2016 census–was used to set the quotas [19]. By its broad representation, the latter was judged as a good proxy for our target population of pregnant women in France. The quotas for mothers of children under 1 year old were used to calculate weightings using Newton's algorithm in order to obtain weighted individual data for the statistical analysis presented herein [20]. Specifically, these quotas comprised age group, socio-professional category, region of residence, size of urban area, and parity.

Eligible women for Covimater (see below) were invited by BVA to answer a web-based questionnaire between 6–20 July 2020, i.e., two months after the end of the first lockdown in France (March to May 2020). The two-month interval was chosen to avoid the memory bias associated with a longer interval. Demographic/socio-economic data, pandemic and lockdown-related data, participants' perceptions of the pandemic, data on their pregnancy, their health, and on pregnancy monitoring during first lockdown were collected.

We compared our sample to another data source (the National Medical and Administrative Database) in order to validate its representativeness. No significant difference in available data for age group, region of residence or parity was observed between the women participating in Covimater and women in the whole French population who gave birth in a hospital maternity ward in France in 2017 (i.e., 99% of pregnant women in France) [21]. Our study shows, with a power of 99%, a difference of at least 20% concerning the variable of interest (see definition below) between two subgroups of balanced/unbalanced women.

### Participants

**Inclusion criteria.**   Women who were pregnant during the first lockdown (17 March–11 May 2020), aged 18 years and over, and residents in metropolitan France.

**Exclusion criteria.**   Women who were pregnant during the lockdown but with limited exposure to it: those who delivered in the first two weeks of the lockdown and those whose first week of gestation began during the last two weeks of the lockdown (deducted from the expected date of delivery reported by the women).

### Issue of interest: Unmet need to communicate with health professionals

Women who reported an unmet need to communicate with a health professional (gynaecologist, nurse, generalist, midwife, etc.) were those who answered "I was not able to discuss this topic with a health professional but I would have liked to", to at least one of six questions regarding: (i) the risk of being infected with SARS-CoV-2 and the possibility of having severe symptoms of COVID-19 disease, (ii) the risk of transmission of SARS-CoV-2 to their baby and the consequences for the latter, (iii) the course of their pregnancy monitoring during the pandemic, (iv) the delivery process in the context of the pandemic, (v) the course of their maternity stay, and (vi) the possibility of breastfeeding without risk to their child during the pandemic.

Women who answered "No, because I did not need it" or "Yes, I met with a health professional" to all six questions were considered to have had their need for communication with health professionals fully satisfied.

### Comparisons

Explanatory variables were divided into five main themes:

Demographic and socio-economic: age, socio-professional category (SPC) reduced into SPC+ (self-employed women, managers, intermediate professions), SPC- (employees, blue-collar workers) and inactive women (students and other professionally inactives), educational level (equal to or higher than secondary school diploma, lower than secondary school diploma), perceived financial situation (comfortable, just getting by, difficult to make ends meet).

Pandemic and lockdown-related: child(ren) under six years of age (*i.e.*, younger than school age in France) in the household during the lockdown, SARS-CoV-2 healthcare system severity as reported by the Ministry of Health on 1 May 2020 in their region of residence (coded as green, orange or red, reflecting increased epidemic pressure on the healthcare system) [22], professional workload (did not work, lighter than/same as usual, heavier than usual), self-perceived social support (from family, friends, etc.; Very good, Good, Little or None), having experienced serious disputes or violence (Very-often/Often, Sometimes/Rarely, Never), having COVID-19 type symptoms, family member or friends diagnosed with COVID-19 or had symptoms suggestive of the disease.

Self-perception of the pandemic during the lockdown: Two different scale-based scores were recorded: one for participants' general level of worry about the pandemic situation in France, and another for their perceived vulnerability to SARS-CoV-2 infection (from 0 (not at all worried/vulnerable) to 10 (very worried/vulnerable)). Two dichotomous variables were then created for 'worry' and 'vulnerability', with 7/10 and 6/10 as the thresholds, respectively, corresponding to the average worry or vulnerability observed (7.0 +/- 0.1 and 6.2 +/- 0.1, respectively).

Pregnancy and health: parity, gestational age at the end of lockdown, childbirth (during or after first lockdown), at least one pre-existing chronic disease, at least one pregnancy-related pathology (see details of pathologies in Table 1), overweight/obesity status before pregnancy (based on Body Mass Index≥25kg/m2; see details in Table 1).

Pregnancy monitoring during first lockdown: had a consultation/examination cancelled/postponed on a health professional's initiative, change in health professional from the one who usually followed them, teleconsultation (video or telephone) for pregnancy monitoring.

### Ethics and endpoint

Covimater received approval from the Saint Maurice Hospital Ethics Committee on 01/07/2020 (approval number n˚2020–1). Internet panel volunteers included in the Covimater study

**Table 1. Description of pregnant women during the first COVID-19-related lockdown (March to May 2020) who participated in the Covimater survey (n = 500), France (July 2020).**

| | N (%) or mean (sd)* | [95%CI**] |
|---|---|---|
| **Demographic and socio-economic characteristics** | | |
| Age (in years) | 31.4 (5.1) | [30.8–31.9] |
| Socio-professional category (SPC)[a] | | |
| SPC + | 192 (38.4) | [33.9–43.2] |
| SPC - | 180 (36.1) | [31.8–40.6] |
| Inactive | 128 (25.5) | [20.5–31.2] |
| Educational level | | |
| Equal to or higher than secondary school diploma | 391 (78.1) | [73.6–82.1] |
| Lower than secondary school diploma | 109 (21.9) | [17.9–26.4] |
| Perceived financial situation | | |
| Comfortable | 246 (49.2) | [44.2–54.2] |
| Just getting by | 159 (31.7) | [27.2–36.6] |
| Difficult to make ends meet | 95 (19.1) | [15.2–23.7] |
| **Pandemic and lockdown related variables** | | |
| Child(ren) under six years of age in the household during the lockdown | 234 (46.8) | [41.8–51.8] |
| SARS-CoV-2 healthcare system severity (colour-coded) for the region of residence[b] | | |
| Green zone | 127 (25.4) | [21.1–30.2] |
| Orange zone | 150 (30.0) | [25.7–34.7] |
| Red zone | 223 (44.6) | [39.7–49.6] |
| Self-perceived social support | | |
| Very good | 180 (36.0) | [31.3–40.9] |
| Good | 231 (46.1) | [41.2–51.1] |
| Little or none | 89 (17.9) | [14.5–21.8] |
| Serious disputes or violence | | |
| Very-often/ Often | 11 (2.3) | [1.10–4.60] |
| Sometimes / Rarely | 129 (25.8) | [21.7–30.4] |
| Never | 360 (71.9) | [67.2–76.2] |
| Having had COVID-19 type symptoms | 92 (18.4) | [14.9–22.6] |
| Family member or friends diagnosed with COVID-19 or had symptoms suggestive of the disease | 171 (34.2) | [29.7–39.0] |
| **Self-perception of the pandemic during first lockdown** | | |
| Perceived a general worry about the SARS-CoV-2 pandemic (max.10; n = 485) > 7/ 10[c] | 234 (48.3) | [43.3–53.3] |
| Perceived vulnerability to severe forms of COVID -19 disease (max. 10; n = 459) >6/10[c] | 250 (54.6) | [49.4–59.6] |
| **Pregnancy and health** | | |
| Primiparous | 203 (40.6) | [35.8–45.6] |
| Gestational age (weeks) at the end of first lockdown[d] | | |
| <10 | 34 (6.8) | [4.70–9.80] |
| 10–20 | 177 (35.4) | [30.8–40.3] |
| 20–30 | 180 (36.1) | [31.4–41.0] |
| 30–40 | 77 (15.4) | [12.1–19.4] |
| > 40 | 32 (6.3) | [4.30–9.20] |
| Childbirth | | |
| During lockdown | 34 (6.8) | [4.70–9.80] |
| After lockdown | 466 (93.2) | [90.2–95.2] |

*(Continued)*

**Table 1.** (Continued)

| | N (%) or mean (sd)[*] | [95%CI[**]] |
|---|---|---|
| Pre-existing Chronic disease(s)[e] | 152 (30.3) | [25.8–35.1] |
| Pregnancy-related pathology(ies)[f] | 119 (23.7) | [19.9–28.0] |
| Overweight/Obesity status before pregnancy[g] | 212 (42.4) | [37.5–47.4] |
| **Pregnancy monitoring during first lockdown** | | |
| Cancelled/postponed pregnancy consultations or examinations at the initiative of a health professional | 182 (36.3) | [31.6–41.3] |
| Forewent/postponed pregnancy consultations or examinations at the initiative of the women[h] | 117 (23.4) | [18.8–27.7] |
| Teleconsultations (video or telephone) for pregnancy monitoring | 197 (39.4) | [34.6–44.4] |
| Change of health professional than the referring professional | 74 (14.9) | [11.7–18.8] |
| Having an unmet need to communicate with health professionals about course of pregnancy/childbirth during pandemic | | |
| No | 295 (59.0) | [53.9–63.8] |
| Yes | 205 (41.0) | [36.1–46.1] |

[*] Weighted and rounded values using Newton's algorithm [20] for discrete or qualitative variables. For continuous variables, mean (standard deviation) were presented.

[**] 95% Confidence Interval

a Women on maternity leave and unemployed women were classified according to their current SPC category or their most recent category prior to ending work, respectively.

b Estimated by the Ministry of Health on 1 May 2020 on the basis of two variables: i) Virus circulation level (i.e., percentage of emergency room admissions for suspected COVID-19) and ii) Strain on hospital intensive care unit capacity (i.e., occupancy rate of intensive care beds by patients with COVID-19), coded as green, orange or red, reflecting increased epidemic pressure on the healthcare system [22].

c Scores for participants' general worry about the pandemic situation and for their perceived vulnerability to SARS-CoV-2 infection during the first lockdown (from 0 (not at all worried/vulnerable) to 10 (very worried/vulnerable)). Two dichotomous 'low/high' variables were then created for 'worry' and 'vulnerability', with 7/10 and 6/10 as the thresholds, respectively (see details in methods). No documented data for 15 and 41 pregnant women in terms of level of worry about the pandemic or level of perceived vulnerability to severe forms of COVID -19, respectively.

d At the end of the first lockdown (11 May 2020) or at the date of childbirth if women gave birth during lockdown.

e Diabetes, Overweight/Obesity status before pregnancy, High Blood Pressure, Asthma, Cardiac condition, Autoimmune disease, mental illness, inherited bleeding disorders.

f Gestational diabetes, pre-eclampsia, preterm labour, gestational hypertension.

g Body Mass Index≥25kg/m2.

h Also includes women who did not start monitoring despite a gestational age of 15 weeks.

were informed by mail of the study's purpose then given the choice to participate in the survey. Only pseudonymised databases were transmitted to Santé publique France. The data are stored on Santé publique France's servers, respecting the agency's data security and confidentiality standards.

## Statistical analysis

A robust variance Poisson regression model was used to estimate unadjusted and adjusted prevalence ratios (aPR) [23] for having had an unmet need to communicate with a health professional. Factors associated with this outcome which had a p-value<0.20 in bivariate analysis

or which were judged clinically relevant (e.g., gestational age at the end of the lockdown period and parity) were introduced into the multivariable model. When several variables were possibly collinear, the model with the best likelihood score (lowest Bayesian Information Criterion) was selected. Fractional polynomials confirmed a linear relationship between continuous variables included in the models and the studied prevalence of the outcome. A manual stepwise descending approach was applied. The final model included all variables independently associated with the variable of interest (p-value<0.05) after epidemiological reflection and according to the clinical relevance of each variable at each step of the procedure. As indicated by Zou, PRs were interpreted in the same way as relative risks [24].

All statistical analyses were performed using Stata® software version 14.2 (Stata Corp., College Station, TX, USA).

## Results

### Characteristics of women included in Covimater (Table 1)

The mean age of the Covimater study sample (n = 500) was 31.4 years (sd = 5.1). The majority (78.1%) had a secondary school diploma or higher level of education, 36.1% were classified SPC-, 25.5% were inactive, 31.7% declared they just got by financially, while 19.1% reported that they could not make ends meet. Among the 500 women in the sample, 40.6% were primiparous. Nearly one in six women (17.9%) received little or no social support during the lockdown, 28% experienced serious arguments and/or a climate of violence, and almost one in two (48.3%) reported having a level of worry higher than 7 (out of 10) about the pandemic during the same first lockdown.

With regard to pregnancy monitoring during the first lockdown, 36.3% reported postponements or cancellations of consultations/examinations by their hospital or health professionals, and 39.4% had had teleconsultations. Furthermore, 14.9% of the women who had started their pregnancy monitoring during the first lockdown declared that they had changed health professionals from the one who usually followed them.

### Pregnant women's unmet need to communicate with a health professional during France's first SARS-CoV-2 pandemic-related lockdown (17 March to 11 May 2020) (Fig 1)

Two in five (41%) participants in Covimater indicated that they had an unmet need to communicate with health professionals on at least one of the six themes studied concerning the SARS-CoV-2 pandemic, pregnancy care and delivery process. The two most frequent themes not discussed with health professionals were i) the risk of transmitting SARS-CoV-2 to their unborn child and the consequences for the latter (29.3%), and ii) the risk of being infected and having severe symptoms (27.4%). Approximately one in six women reported that they would have liked to have been able to talk with a professional about the delivery process (16.1%), the maternity stay (16.4%), and the possibility of breastfeeding during the SARS-CoV-2 pandemic (15.3%).

Proportion of women with an unmet need to communicate with a health professional. Among the 500 pregnant women during the lockdown, 102 had given birth at the time of completing the web-questionnaire.

### Factors associated with unmet need to communicate with a health professional during the first SARS-CoV-2 pandemic lockdown (Table 2)

The following subgroups had a significantly higher prevalence of having at least one unmet need for information for the six themes addressed in the study: women who were

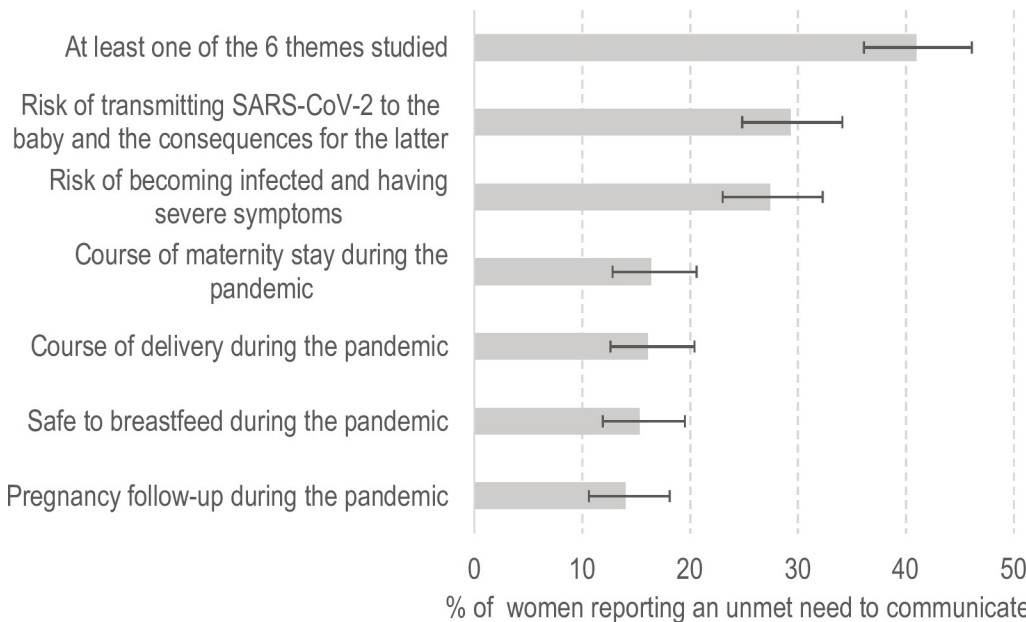

**Fig 1. Pregnant women's need to communicate with a health professional during the SARS-CoV-2 pandemic lockdown in France.**

professionally inactive (aPR = 1.58, CI95%[(1.14–2.21)]), those with a level of education lower than secondary school diploma (1.38, [1.05–1.81]), those having experienced often or very-often violence or serious arguments during the lockdown (2.12, [1.28–3.52]), those very worried about the pandemic (1.41, [1.11–1.78]), those who were primiparous (1.36, [1.06–1.74]) and those who had a pregnancy consultation cancelled/postponed at the initiative of health professionals during the lockdown (1.35, [1.06–1.73]).

## Discussion

Nearly 41% of pregnant women in Covimater reported that they had tried in vain to communicate with a health professional on at least one of the six topics studied in the survey. The topics that pregnant women would have liked to discuss were mainly related to the risk and complications for them and their unborn/born baby in the event of SARS-CoV-2 infection, and the impact of the pandemic on their pregnancy monitoring or delivery. Primiparous women, those professionally inactive, those with an educational level below secondary school diploma, those who had experienced violence during lockdown, those very worried about the pandemic, and finally, those whose pregnancy consultations had been postponed or cancelled at the initiative of a health professional were all more likely to have had an unmet need to communicate with a healthcare professional.

Health literacy is the degree to which individuals have the capacity to obtain, process, and understand basic health information needed to make appropriate health decisions [25].

Greater health literacy is associated with better health [26]. One of the most important determinants of health literacy is education level [26]. According to our results, pregnant women who did not have a secondary school diploma were more likely to report an unmet need to communicate with a health professional. Because of poor health literacy, these women may have had more difficulties to express their need and therefore to obtain answers to their

**Table 2. Factors associated with an unmet need to communicate with a healthcare professional about course of pregnancy or childbirth during the first SARS-CoV-2 pandemic lockdown, Covimater survey (n = 500), France (July 2020).**

| | N (%) or mean (sd)* | Adjusted PR [95% CI] ** | p-value** |
|---|---|---|---|
| Age (in years) | 30.6 (5.4) | 0.98 [0.96–1.00] | 0.137 |
| Gestational age (in weeks)[a] | 23.4 (8.9) | 0.99 [0.98–1.01] | 0.477 |
| Socio-professional category[b] | | | |
| SPC+ | 65 (33.8) | 1 | |
| SPC- | 74 (41.1) | 1.05 [0.79–1.40] | 0.716 |
| Inactive | 66 (51.5) | **1.58 [1.14–2.21]** | **0.007** |
| Parity | | | |
| Primiparous | 93 (45.8) | **1.36 [1.06–1.74]** | **0.014** |
| Multiparous | 112 (37.7) | 1 | |
| Educational level | | | |
| Equal to or higher than secondary school diploma | 145 (37.1) | 1 | |
| Lower than secondary school diploma | 60 (55.0) | **1.38 [1.05–1.81]** | **0.022** |
| Serious disputes or violence during the lockdown | | | |
| Never | 137 (38.0) | 1 | |
| Sometimes / Rarely | 61 (47.3) | **1.29 [1.02–1.63]** | **0.033** |
| Very-often/ Often | 7 (63.6) | **2.12 [1.28–3.52]** | **0.004** |
| Self-perceived general worry about the SARS-CoV-2 pandemic (max.10)[c] | | | |
| Score less than or equal to 7 | 90 (35.8) | 1 | |
| Score above 7 | 113 (48.3) | **1.41 [1.11–1.78]** | **0.004** |
| Cancelled/postponed pregnancy consultations or examinations at the initiative of a health professional | | | |
| No | 116 (36.5) | 1 | |
| Yes | 89 (48.9) | **1.35 [1.06–1.73]** | **0.016** |

* Weighted and rounded values using Newton's algorithm [20] for discrete or qualitative variables. For continuous variables, mean (standard deviation) were presented.

** Adjusted Prevalence Ratio (aPR), 95%Confidence Interval (95%CI) and p-value obtained with robust variance Poisson regression model.

a At the end of the first French lockdown (11 May 2020) or at the date of childbirth if women gave birth during lockdown.

b Women on maternity leave and unemployed women were classified according to their current SPC category or their most recent category prior to ending work respectively.

c 15 women did not document their general worry score whose 2 with an unmet need to communicate with a healthcare professional.

questions when they talked to the health professionals; accordingly, they were more likely to be dissatisfied.

In Covimater, primiparous women were more likely to attempt to communicate with health professionals, particularly concerning the process of delivery and their stay in the maternity ward. On this subject, the National Perinatal Survey (NPS) carried out in France in all maternity units in 2016 reported that birth and parenthood preparation sessions were more often attended by primiparous women [27]. Primiparous women may have been more likely to have unmet need to communicate with a healthcare professional than their multiparous counterparts.

In our analyses, postponements/cancellations of consultations/examinations by health professionals were associated with a higher likelihood of reporting an unmet need to communicate with healthcare professionals during France's first lockdown. Several studies have underlined the importance of the patient/caregiver relationship in medical monitoring (compliance with care, treatment, and health examinations). It is therefore crucial, even outside the context of the pandemic and lockdown, to maintain personalised monitoring of pregnant women as much as possible [28].

In Covimater, pregnant women who were very worried about the pandemic in general were more likely to have had an unmet need to communicate with a health professional during the first lockdown. The self-reported reasons for worry were mainly related to their pregnancy, the risk of infection of their vulnerable relatives, the risk of being infected themselves and of transmitting the virus to their unborn child (article submitted for publication). In our analysis, the topics which women would have liked to talk about but could not overlapped with some of these most frequent reasons for worry: the risk of being infected and having a severe form of COVID-19 disease; the risk of transmitting the virus to their unborn child, and the consequences for the latter, the course of the delivery, and the stay in the maternity ward. The need to communicate with health professionals on these subjects reflects the initial worldwide uncertainty about the consequences of infection and COVID-19 disease for pregnant women throughout the medical and scientific community during the first international lockdowns [3, 6, 7, 9].

Our results showed an association between having experienced violence during the first lockdown and pregnant women's unmet need to communicate with health professionals. Although the percentage of pregnant women that reported violence or serious arguments during the first lockdown was high (28.1%), it was significantly lower than that obtained for women of childbearing age (18–49 years) in the CoviPrev study (32.9%, p = 0.03), a repeated cross-sectional study in the French general population using the same methodology and conducted at the same time as Covimater [29]. Some studies but not all reported a higher risk of violence in pregnant women than in non-pregnant women [30, 31]. With regard to prenatal care, several studies have shown that pregnant women experiencing domestic violence were more likely to delay antenatal visits; for some, this was because partners prevented or discouraged them from having visits, while others felt embarrassed about the possibility of being judged in public because of obvious signs of violence [32, 33]. A study conducted in the United States showed that pregnant women who experienced domestic violence (*vs* those who did not) were 1.8 times more likely (CI95%[1.5–2.1]) to delay entry into prenatal care [32]. This may partly explain why, in our analysis, pregnant women who experienced violence during France's first lockdown were more likely to have an unmet need for communication with health professionals. Efforts to detect violence against women at an early stage of pregnancy should be continued to prevent its harmful impact on health.

Health professionals are not the only sources of information for the general population; many health promotion campaigns have demonstrated their impact in recent years [34–37]. One example is the 'Antibiotics are not automatic' campaign conducted in France in 2002. This slogan was strongly featured on social networks, television and newspapers, and halved the number of false responses in surveys concerning the use of antibiotics. In addition, this campaign created a need for information and knowledge among patients; more specifically, 55% of the participants felt that the campaign made them want to know more [35]. In May 2020, during the COVID-19 pandemic in France, the following message targeting patients and vulnerable populations including newborns and pregnant women was broadcast on television and radio: "During the pandemic, whatever your health problem, make sure you get care" [38]. It is essential to continue to use these communication channels to inform and reassure populations during a health crisis.

Our work therefore highlighted the importance of maintaining or promoting communication between health professionals and pregnant women during the pandemic. Specific information campaigns could also be enhanced and circulated through diverse media channels (e.g., radio or television) to help reassure this population during the pandemic.

In the literature, little information was available on patients' needs for communication with health professionals, even less in the context of the ongoing SARS-CoV-2 pandemic. Covimater brought new insight to this important topic and identifies specific groups of women who

should be targeted by public policies, which constitutes the study's primary strength. Secondly, Covimater included women with different gestational ages, unlike studies from other countries that mostly focus on the third trimester of pregnancy during the current pandemic. Further-more, Covimater succeeded in identifying significant associations with the variable of interest despite the fact that some of the groups compared were unbalanced in size (with consequently reduced power).

Covimater also had some limitations. First, the use of a panel of volunteers and quota sam-pling could imply an inclusion bias in the pregnant women who accepted to participate for the survey. However, no alternative method available would have permitted the study to take place in sufficient time to avoid a significant recall bias. The further away the lockdown was, the more difficult it would have been to collect reliable information from women about their behaviour and feelings during the period. Consequently, greater caution is required when interpreting the statistical inference of our results than would be needed for random sample studies. Second, sampling bias could explain the poor estimation of the percentage of pregnant women with pre-existing chronic diseases like diabetes or obesity (1.5 *vs* 0.5% and 2.4 *vs* 12% in Covimater *vs* NPS study, respectively). Third, as the study questionnaire was self-adminis-tered, there is a risk that respondents misinterpreted questions, as well as a risk of recall biases or potential social desirability. However, there is no reason why any of the study's above-men-tioned limitations should only affect a particular sub-group of pregnant women.

## Conclusions

During the first SARS-CoV-2 pandemic lockdown in France, a high proportion of pregnant women declared an unmet need to communicate with a health professional. The Covimater study made it possible to identify pregnant women who were at particular risk of this unmet need, with a special focus on women who were victims of violence. Our results underline the importance of public policies aimed at preventing this communication deficit, for example, by promoting access for pregnant women to healthcare/patient communication channels and by increasing the availability of information on the different types of media used by these women.

## Acknowledgments

Our thanks to Dorothée Lamarche (BVA group) for her invaluable help in creating the study questionnaire, and to Jude Sweeney (Milan, Italy) for the English revision and editing of this manuscript.

## Author Contributions

**Conceptualization:** Lucia Araujo-Chaveron, Alexandra Doncarli.

**Formal analysis:** Lucia Araujo-Chaveron, Alexandra Doncarli.

**Methodology:** Lucia Araujo-Chaveron, Alexandra Doncarli.

**Validation:** Catherine Crenn-Hebert, Virginie Demiguel, Julie Boudet-Berquier, Yaya Barry, Maria-Eugênia Gomes Do Espirito Santo, Andréa Guajardo-Villar, Claudie Menguy, Anouk Tabaï, Karine Wyndels, Alexandra Benachi, Nolwenn Regnault.

**Writing – original draft:** Lucia Araujo-Chaveron, Alexandra Doncarli.

**Writing – review & editing:** Catherine Crenn-Hebert, Virginie Demiguel, Julie Boudet-Ber-quier, Yaya Barry, Maria-Eugênia Gomes Do Espirito Santo, Andréa Guajardo-Villar, Clau-die Menguy, Anouk Tabaï, Karine Wyndels, Alexandra Benachi, Nolwenn Regnault.

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
