## [Decision Letter · Decision Letter 0]

29 Oct 2021

PONE-D-21-31849Pregnant women's unmet need to communicate with a health professional during the SARS-CoV-2 pandemic in France: the Covimater cross-sectional studyPLOS ONE

Dear Dr. Doncarli,

Thank you for submitting your manuscript to PLOS ONE. After careful consideration, we feel that it has merit but does not fully meet PLOS ONE’s publication criteria as it currently stands. Therefore, we invite you to submit a revised version of the manuscript that addresses the points raised during the review process.

We look forward to receiving your revised manuscript.

Kind regards,

Gabriel O Dida, PhD

Academic Editor

PLOS ONE

Journal Requirements:

3. We note that you have stated that you will provide repository information for your data at acceptance. Should your manuscript be accepted for publication, we will hold it until you provide the relevant accession numbers or DOIs necessary to access your data. If you wish to make changes to your Data Availability statement, please describe these changes in your cover letter and we will update your Data Availability statement to reflect the information you provide

Additional Editor Comments (if provided):

Consider only relevant suggested reference sources

Reviewers' comments:

Reviewer's Responses to Questions

**Comments to the Author**

1. Is the manuscript technically sound, and do the data support the conclusions?

Reviewer #1: Yes

Reviewer #2: Partly

2. Has the statistical analysis been performed appropriately and rigorously? 

Reviewer #1: Yes

Reviewer #2: Yes

3. Have the authors made all data underlying the findings in their manuscript fully available?

Reviewer #1: Yes

Reviewer #2: Yes

4. Is the manuscript presented in an intelligible fashion and written in standard English?

Reviewer #1: Yes

Reviewer #2: Yes

5. Review Comments to the Author

Reviewer #1: I am glad to assess this study entitled, " Pregnant women's unmet need to communicate with a health professional during the SARS-CoV-2 pandemic in France: the Covimater cross-sectional study."

I suggest some minor corrections to check the typo errors in writing to enhance the English quality to reach the scientific merit for the publication of this study. I will this article after the following modifications.

This present study explains that

When the pandemic started, healthcare systems throughout the world rapidly reorganised to respond to what was an unprecedented situation. We aimed to i) describe the unmet need of pregnant women living in France to communicate with health professionals about the pandemic and their pregnancy, ii) assess the socio demographic, medical and contextual factors associated with this unmet need I am in favor of this study and will recommend for publication. However, the authors need to revise the manuscript and work according to my suggestions to enhance the quality. I will accept this paper for publication after these minor changes as suggested below.

Introduction and literation sections

I recommend the authors add suggested articles in the introduction and literature sections. These research articles have identified health-related topics I believe it will improve the quality of your work. I strongly suggested them to improve this section a bit more. I advise authors to revisit their introduction and literature sections of the recommended studies and cite these studies to enhance your research study's quality to reach scientific merit for publication.

Wang, C., Wang, D., Abbas, J., Duan, K., & Mubeen, R. (2021). Global financial crisis, smart lockdown strategies, and the COVID-19 spillover impacts: A global perspective implications from Southeast Asia. Front Psychiatry, 12, 1-14. doi:10.3389/fpsyt.2021.643783

Abbas, J., Hussain, I., Hussain, S., Akram, S., Shaheen, I., & Niu, B. (2019). The Impact of Knowledge Sharing and Innovation upon Sustainable Performance in Islamic Banks: A Mediation Analysis through an SEM Approach. Sustainability, 11(15), 4049. doi:10.3390/su11154049

NeJhaddadgar, N., Ziapour, A., Zakkipour, G., Abolfathi, M., & Shabani, M. (2020, Nov 13). Effectiveness of telephone-based screening and triage during COVID-19 outbreak in the promoted primary healthcare system: a case study in Ardabil province, Iran. Z Gesundh Wiss, 1-6. https://doi.org/10.1007/s10389-020-01407-8

Literature

I want to see publish this creative study after some corrections. I have endorsed this study as; it deserves the merit for publication. However, I suggest the authors make minor corrections according to my advice. Please read the suggested studies and cite them in the introduction, literature, and method sections. How corporate social responsibility, innovation and social media and internet use is helpful. Add few lines in the introduction and literature sections. How companies are practicing CSR, business, entrepreneurial networks with innovation and knowledge sharing to improve the business performance and provide better healthcare medicines?

Abbas, J., Raza, S., Nurunnabi, M., Minai, M. S., & Bano, S. (2019). The Impact of Entrepreneurial Business Networks on Firms’ Performance Through a Mediating Role of Dynamic Capabilities. Sustainability, 11(11), 3006. doi:10.3390/su11113006

Azizi, M. R., Atlasi, R., Ziapour, & Naemi, R. (2021). Innovative human resource management strategies during the COVID-19 pandemic: A systematic narrative review approach. Heliyon, 7(6), e07233. doi:10.1016/j.heliyon.2021.e07233

Abbas, J., Zhang, Q., Hussain, I., Akram, S., Afaq, A., & Shad, M. A. (2020). Sustainable Innovation in Small Medium Enterprises: The Impact of Knowledge Management on Organizational Innovation through a Mediation Analysis by Using SEM Approach. Sustainability, 12(6), 2407. doi:https://doi.org/10.3390/su12062407

Azadi, N. A., Ziapour, A., Lebni, J. Y., Irandoost, S. F., & Chaboksavar, F. (2021). The effect of education based on health belief model on promoting preventive behaviors of hypertensive disease in staff of the Iran University of Medical Sciences. Archives of Public Health, 79(1), 69. doi:10.1186/s13690-021-00594-4

Materials and Methods

The results section of the paper presents a good view of the study. This work presents a notable investigation on a selected topic. I suggest the authors to present high quality graphs. By including some graphical presentations will improve the quality of this study. Please see the proposed studies and see the graphical representation. Improve your work like these studies and cite them in this section.

Paulson, K. R., Kamath, A. M., Alam, T., Bienhoff, K., Abady, G. G., . . . Kassebaum, N. J. (2021). Global, regional, and national progress towards Sustainable Development Goal 3.2 for neonatal and child health: all-cause and cause-specific mortality findings from the Global Burden of Disease Study 2019. The Lancet, 1-36. doi:10.1016/s0140-6736(21)01207-1

Hussain, T., Wei, Z., & Nurunnabi, M. (2019). The Effect of Sustainable Urban Planning and Slum Disamenity on The Value of Neighboring Residential Property: Application of The Hedonic Pricing Model in Rent Price Appraisal. Sustainability, 11(4), 1144. doi:10.3390/su11041144

Abbas, J., Aman, J., Nurunnabi, M., & Bano, S. (2019). The Impact of Social Media on Learning Behavior for Sustainable Education: Evidence of Students from Selected Universities in Pakistan. Sustainability, 11(6). https://doi.org/10.3390/su11061683

Mubeen, R., Han, D., Abbas, J., & Hussain, I. (2020). The Effects of Market Competition, Capital Structure, and CEO Duality on Firm Performance: A Mediation Analysis by Incorporating the GMM Model Technique. Sustainability, 12(8), 3480. doi:10.3390/su12083480

Discussion

I suggest the authors to discuss the effects of the COVID-19. I suggest you to cite these studies. Read the proposed studies to improve your results and discussion section. See the recommended studies and improve your sections. How companies are practicing CSR, business, entrepreneurial networks with innovation and knowledge sharing to improve the business performance and provide better performance?

Su, Z., McDonnell, D., Wen, J., Kozak, M., Šegalo, S., . . . Xiang, Y.-T. (2021). Mental health consequences of COVID-19 media coverage: the need for effective crisis communication practices. Globalization and Health, 17(1), 4. doi:10.1186/s12992-020-00654-4

Abbas, J., Mahmood, S., Ali, H., Ali Raza, M., Ali, G., Aman, J., . . . Nurunnabi, M. (2019). The Effects of Corporate Social Responsibility Practices and Environmental Factors through a Moderating Role of Social Media Marketing on Sustainable Performance of Firms’ Operating in Multan, Pakistan. Sustainability, 11(12), 3434. doi:10.3390/su11123434

Aqeel, M., Shuja, K. H., Rehna, T., Ziapour, A., Yousaf, I., & Karamat, T. (2021). The Influence of Illness Perception, Anxiety and Depression Disorders on Students Mental Health during COVID-19 Outbreak in Pakistan: A Web-Based Cross-Sectional Survey. International Journal of Human Rights in Healthcare, 14, 1-14.

Abbas, J. (2020). The Impact of Coronavirus (SARS-CoV2) Epidemic on Individuals Mental Health: The Protective Measures of Pakistan in Managing and Sustaining Transmissible Disease. Psychiatr Danub, 32(3-4), 472-477. https://doi.org/10.24869/psyd.2020.472

Conclusion

I suggest you make a separate heading of the conclusion and do not mix it with implications.

Policy Recommendations

I again recommend you to make a separate heading of the Policy Recommendations.

The conclusion section is acceptable. Overall, this presents a good piece of research work. I recommend that authors do a little more work and revise this article accordingly. I suggest the authors check English quality and fix some weak sentences. If you have already taken English editing service, ask them to recheck the quality to meet scientific merit for publication. I endorse this manuscript for publication after minor corrections, as suggested.

Abbasi, K. R., Abbas, J., & Tufail, M. (2021). Revisiting electricity consumption, price, and real GDP: A modified sectoral level analysis from Pakistan. Energy Policy, 149, 112087. doi:10.1016/j.enpol.2020.112087

Local Burden of Disease, H. I. V. C. (2021, 2021/01/08). Mapping subnational HIV mortality in six Latin American countries with incomplete vital registration systems. BMC Medicine, 19(1), 4. https://doi.org/10.1186/s12916-020-01876-4

Pay attention of English quality to reach scientific merit. I suggest to cite these six studies to improve the quality of the Introduction, Literature and Methods sections. I accept and endorse this manuscript for publication after the suggested minor corrections.

Reviewer #2: Abstract:

- Add a short background explaining the study question.

- The period of the study is confusing "July 2020" and "March to May 2020". Clarify

- Add detailed data about the participants and their criteria.

- The conclusion should be precise. Add the future directions.

Introduction:

- Define "SARS-COV-2 and its negative effects on pregnant women" in detail.

- Explain the measured variables.

- The significance of the study needs more details. What will the study add to the knowledge?

- Add the study hypothesis.

Methods:

- The study design, ethics, and setting are not clear.

- How and who administrates the data collection?

- How did you achieve the validity and reliability of the outcome measures?

- Please, reframe the components (SPICES) for methods

i. Study design, setting, sample size

ii. Participant (inclusion and exclusion criteria"

iii. Issue of interest (exposure)

iv. Comparisons

v. Ethics and endpoint

vi. Statistical analysis

- Mention the settings and locations where the data were collected.

- How was the sample size determined?

- Who enrolled participants?

- Who assigned participants?

Discussion:

- The strengths should be demonstrated in detail.

- Add your recommendations with study implications.

- The main limitation of the study is not demonstrated.

6. PLOS authors have the option to publish the peer review history of their article (what does this mean?). If published, this will include your full peer review and any attached files.

Reviewer #1: No

Reviewer #2: **Yes: **Walid Kamal Abdelbasset

---

## [Author Response · Author response to Decision Letter 0]

19 Feb 2022

Dear reviewers,

Thank you very much for all the questions you have asked, highlighting the need for clarification and also helping to improve this manuscript.

Unfortunately, we submit our revisions to the comments of Reviewer 2 only (see attached document) as we assume that the remarks of Reviewer 1 correspond to another article.

Yours sincerely,

Alexandra Doncarli for the co-authors.

---

## [Decision Letter · Decision Letter 1]

1 Apr 2022

Pregnant women's unmet need to communicate with a health professional during the SARS-CoV-2 pandemic in France: the Covimater cross-sectional study

PONE-D-21-31849R1

Dear Dr. Doncarli,

We’re pleased to inform you that your manuscript has been judged scientifically suitable for publication and will be formally accepted for publication once it meets all outstanding technical requirements.

Kind regards,

Gabriel O Dida, PhD

Academic Editor

PLOS ONE

Additional Editor Comments (optional):

Reviewers' comments:

Reviewer's Responses to Questions

**Comments to the Author**

1. If the authors have adequately addressed your comments raised in a previous round of review and you feel that this manuscript is now acceptable for publication, you may indicate that here to bypass the “Comments to the Author” section, enter your conflict of interest statement in the “Confidential to Editor” section, and submit your "Accept" recommendation.

Reviewer #2: All comments have been addressed

Reviewer #3: (No Response)

2. Is the manuscript technically sound, and do the data support the conclusions?

Reviewer #2: Yes

Reviewer #3: (No Response)

3. Has the statistical analysis been performed appropriately and rigorously? 

Reviewer #2: Yes

Reviewer #3: (No Response)

4. Have the authors made all data underlying the findings in their manuscript fully available?

Reviewer #2: Yes

Reviewer #3: (No Response)

5. Is the manuscript presented in an intelligible fashion and written in standard English?

Reviewer #2: Yes

Reviewer #3: (No Response)

6. Review Comments to the Author

Reviewer #2: Appreciating the authors for their responses to my previous comments. All comments have been addressed. Congrats

Reviewer #3: (No Response)

7. PLOS authors have the option to publish the peer review history of their article (what does this mean?). If published, this will include your full peer review and any attached files.

Reviewer #2: **Yes: **Walid Kamal Abdelbasset

Reviewer #3: **Yes: **Navid Rabiee

---

## [Editor Report · Acceptance letter]

7 Apr 2022

PONE-D-21-31849R1 

Pregnant women's unmet need to communicate with a health professional during the SARS-CoV-2 pandemic lockdown in France: the Covimater cross-sectional study 

Dear Dr. Doncarli:

I'm pleased to inform you that your manuscript has been deemed suitable for publication in PLOS ONE. Congratulations! Your manuscript is now with our production department. 

Kind regards, 

on behalf of

Dr. Gabriel O Dida 

Academic Editor

PLOS ONE